# Neurodevelopmental Model Explaining Associations between Sex Hormones, Personality, and Eating Pathology

**DOI:** 10.3390/brainsci13060859

**Published:** 2023-05-25

**Authors:** Ziyu Zhao, Kyle Gobrogge

**Affiliations:** 1Department of Psychological & Brain Sciences, Boston University, Boston, MA 02215, USA; gobrogge@bu.edu; 2Undergraduate Program in Neuroscience, College of Art & Sciences, Boston University, Boston, MA 02215, USA

**Keywords:** hormones, personality, eating disorder, mesocorticolimbic dopamine system, mediation, moderation, gonadal hormones, estrogen/estradiol, testosterone, androgen, disordered eating, eating pathology, binge eating, emotional eating

## Abstract

Clinical scientists have been investigating the relationships between sex hormones, personality, and eating disorders for decades. However, there is a lack of direct research that addresses whether personality mediates or moderates the relationships between sex hormones and eating pathology. Moreover, the neural mechanisms that underlie the interactive associations between these variables remain unclear. This review aims to summarize the associations between these constructs, describe a neural mechanism mediating these relationships, and offer clinical strategies for the early identification and intervention of eating disorders. The gathered evidence shows that aggressiveness, impulsivity, and obsessive-compulsiveness may mediate or moderate the relationships between sex hormones and eating pathology, but only among females. Furthermore, sex hormone receptor density in the mesocorticolimbic dopamine pathway may explain the neural mechanism of these associations. Future research should use more comprehensive personality measurements and assess the mediation and moderation effects of temperament while taking the hormone levels of women across menstrual cycles into account. Additionally, electroencephalography and functional magnetic resonance imaging should be implemented to directly assess brain activity and corroborate these findings.

## 1. Introduction

In 1970, P.J.V. Beaumont studied male patients with anorexia nervosa (AN) and found that they had decreased testosterone levels, potentially due to abnormal pituitary gonadotrophin function [1]. Modern research has since revealed significant effects of hormones on eating pathology. A recent review by Culbert et al. showed that perinatal testosterone exposure offers preliminary protection against eating pathology in neurotypical developing males, but its absence in females increases their risk for disordered eating attitudes and behaviors [2].

Personality is significantly correlated with eating pathology. Research in the 1980s by Michael Strober identified obsessionality as a key personality feature in young female anorexic patients, while impulsivity was associated with bulimic behaviors characterized by binge eating and purging [3]. The DSM-V has better classified eating disorders (EDs), and additional personalities such as perfectionism, impulsiveness, and compulsiveness have been found to be correlated with other eating pathologies [4].

Functional magnetic resonance imaging (fMRI) has been used to investigate the neural correlates of the association between sex hormones and personality. Gray’s reinforcement sensitivity theory [5], which proposes the biological difference between approach and avoidant behaviors, has been widely studied. The revised model includes three major neurobiological networks: the Behavioral Approach System (BAS), Fight-Flight-Freeze System (FFFS), and Behavioral Inhibition System (BIS). Neural activation patterns of BAS are associated with sensitivity to reward—recruiting the ventral tegmental area (VTA), ventral striatum (VS), and prefrontal cortex (PFC) [6]. High FFFS is more likely to activate the hypothalamus and amygdala (AMY)—limbic areas programming avoidant behavior [6]. High BIS is likely to lead to increased activity in the hypothalamus and AMY, while low BIS tends to activate the posterior cingulate cortex (PCC) and dorsal prefrontal cortex (dPFC)—underlying impulse control, arousal and attention [6].

BAS is linked to sensitivity to reward, associating BAS with personality traits related to reward seeking. Research has shown that BAS controls novelty seeking and impulsivity [7,8]. This neurobiological evidence of personality suggests that identifying the neural circuits with a high density of steroid hormone receptors could reveal novel neural mechanisms that explain the relationships between hormones, personality, and eating pathology. Higher sex hormone receptor density could theoretically change how neurons function in the brain with regards to steroid sensitivity. In other words, neural circuits with high steroid hormone receptor density might affect the strength of the connections between steroid hormone neurophysiology and the personalities associated with eating pathology.

This review seeks to answer two questions: (1). Does personality affect the relationship between sex hormones and eating pathology? and (2). What are the neural mechanisms behind the connection between steroid hormones, personality, and eating pathology? Past research has shown a strong association between sex hormones and personality [9,10]. When combining this with evidence of associations between sex hormones and eating pathology, it is possible to investigate whether interactions between neurophysiology and personality contribute to the development of eating pathology. Additionally, this review examines the density of hormone receptors in specific brain areas related to traits associated with disordered eating behaviors. Overall, this review surveys research on interactions between sex hormones, personality, and eating behaviors in both humans and laboratory animals. A neurodevelopmental model is proposed to explain disordered eating attitudes and behaviors, which can inform early clinical identification and treatment of eating pathology.

## 2. Material and Methods

### Literature Search and Criteria

Research was selected via databases PsycINFO (Washington, DC, USA), PsycArticles (Washington, DC, USA), Wiley Library (Hoboken, NJ, USA), and PubMed (Bethesda, MD, USA). The publication time frame was 1980 (introduction of DSM-III) to 2021, and restricted to English, peer-reviewed journals. The searched keywords and procedures included sex OR sex hormones, sex effect, men/women, boys/girls, personality, OR personality traits, and eating disorders OR eating pathology OR anorexia nervosa OR bulimia nervosa OR binge-eating disorder. Articles on the association between sex hormones and personality were included if they addressed the hormonal effect on personality traits or the development of personality disorders. Articles on personality traits and eating pathology were included if they investigated personality traits or disorders as the etiology or comorbidity of eating pathology rather than the outcome. Studies that directly focused on personality’s mediation or moderation effect on the association between sex hormones and eating pathology were also included. It is worth noting that studies focused on ED symptoms rather than clinical EDs, given that hormone levels are altered during the ill state of EDs.

## 3. Discussion

### 3.1. Association between Sex Hormones and Eating Pathology

#### 3.1.1. Androgen

##### Animal Models

There has been limited research on the effects of testosterone on eating behaviors in laboratory animals. Some studies have suggested that removing testosterone from rodents increases sucrose consumption [11], while administering testosterone to female rats leads to male-like eating patterns such as increased food intake [12]. Recent evidence also suggests that perinatal exposure to testosterone can alter post-pubertal eating behaviors in female rats, resulting in a lower preference for palatable foods and an increased propensity for binge eating [13]. Moreover, prenatal exposure to testosterone among male rats can have an organizational effect that drives the development of male-typical physiological and behavioral characteristics, including eating behaviors [14]. On the other hand, the absence of prenatal exposure to testosterone among female rats allows for the development of their female-typical physiological and behavioral characteristics [15,16].

##### Human Models

In human studies, researchers have investigated the binge-eating behaviors of females born with male co-twins, as it is believed that these women are exposed to high levels of testosterone in the womb and are, therefore, expected to have lower binge-eating rates [17] (See Table 1). Another measure used to indirectly assess prenatal testosterone exposure is the ratio of the length of the index finger to that of the ring finger (known as the 2D:4D ratio) [18]. However, the results of studies using this measure have been mixed. Some studies suggest that high prenatal testosterone exposure is associated with low rates of bulimia nervosa (BN) and high rates of AN [19,20], while others show no effect [21,22]. Mikhail and colleagues have suggested that these mixed results may be due to factors such as later testosterone exposure, total testosterone levels, and genetic and environmental risk [23].

On the other hand, the impact of testosterone exposure on binge eating during other neurodevelopmental stages contradicts the protective effect of prenatal testosterone exposure among females. Sundblad et al. (1994) found that among women, high levels of testosterone were linked to BN [25]. Additionally, adult women with polycystic ovary syndrome (PCOS), a condition characterized by increased testosterone levels, were more likely to develop BN [28]. However, some researchers argue that this scenario may be caused by decreased estradiol levels among females with PCOS [46], which could increase their binge-eating frequency [27]. Nevertheless, more research is needed to uncover the underlying mechanism of binge eating among females with PCOS.

Males have a lower frequency to engage in binge eating compared to females (1 in 8 compared to 1 in 2) [27], possibly because they naturally experience prenatal exposure to testosterone. This exposure to testosterone acts as a protective factor against eating pathology, and continued exposure during puberty and adulthood may offer additional protective effects to males compared to females. Similar to findings in animal models, research has shown that high levels of circulating testosterone are associated with low levels of ED symptoms (such as binge eating) among adolescent males, even after adjusting for factors such as age and mood symptoms [26]. This indicates an inverse relationship between testosterone and eating pathology. Another study conducted among adults also found the same inverse relationship between testosterone and eating pathology [29]. Although no data suggest that testosterone is associated with specific EDs, existing evidence suggests that testosterone levels are inversely associated with binge-eating symptoms Table 1.

#### 3.1.2. Ovarian Hormones

##### Animal Models

Research on animal models has demonstrated that removing ovarian hormones in female adult rats and mice increased their binge-eating behavior [32,34], while administering estradiol reduced their food intake [47]. Additionally, a study found that estradiol had a strong effect in reducing binge-eating propensity even in rats exposed to high-risk environments such as food restriction and stress [44]. The inhibitory effect of estradiol on eating was mediated through the establishment of anorexic agent peptides in the brains of both male and female rats [31,35,42].

##### Human Models

Aggregate data suggest that ovarian hormones play a role in females’ higher propensity for binge eating and preference for carbohydrate and fat-dense foods [16]. For example, women in the perimenopause stage with low estradiol levels may be more susceptible to eating pathologies due to the strong inverse relationship between estradiol and ED risk [41]. Additionally, estradiol and progesterone have opposite effects on other ED cognition, such as “drive for thinness” and “body dissatisfaction” [48]. Estradiol has also been shown to moderate the genetic effect on eating pathology among pubescent girls, with stronger effects observed among girls with lower levels of estradiol (~70%) relative to high (~2%) [49]. However, the phenotypic associations between ovarian hormones and binge eating among females do not appear until later neurodevelopmental stages due to the organizational effect of sex hormones [50]. Moreover, it seems that a lower level of estrogen combined with a higher level of progesterone consistently contributes to the risk of eating pathology among women. Studies focusing on female eating behaviors across menstrual cycles have found that participants had peak food intake [51,52], increased incidence of binge eating [36], and emotional eating during the postovulatory stage when estrogen levels are low and progesterone levels are high. The combination of low estradiol and high progesterone also contributed to binge eating among women with BN [33].

Recent research has focused on how the combination of estradiol and progesterone affects symptoms of EDs, particularly binge eating. Studies consistently show that participants have the highest risk of EDs when both hormones peak. This confirms earlier findings that indicate increased incidences of binge eating during the postovulatory phase [36,53]. Although initial research suggested that estradiol had a positive association with disordered eating [30], subsequent studies have found that during the preovulatory phase, when estradiol levels are highest and progesterone levels are lowest, women have the lowest binge-eating risk [33,36,53]. However, the combined effect may be attributed to the antagonizing effect of progesterone on estrogen [40], resulting in lower levels of estrogen and binge-eating symptoms among females. In summary, estradiol might protect against binge-eating symptoms among women, while progesterone might aggravate them Table 1. Few studies have investigated the effect of ovarian hormones on ED symptoms in males.

Previous studies have mainly focused on the link between ovarian hormones and binge-eating or general eating attitudes and behaviors, but there is a gap in understanding how these hormones are related to AN. Young [54] has proposed that irregular responses to estrogen, possibly caused by various biological and physiological factors, may contribute to AN. It is also worth investigating whether a lower level of estradiol is linked to a higher level of restrictive eating, although this hypothesis contradicts past evidence. However, it is challenging to identify a specific neuromechanism that is predominantly responsible for AN [54] because women with AN in their ill state have dysregulated physiological levels of ovarian hormones [55]. Therefore, more studies are needed to explore the relationship between estradiol and restrictive eating in patient populations.

### 3.2. Association between Personality and Eating Pathology

Research suggests that several personality traits are associated with eating pathologies, such as perfectionism, obsessive-compulsiveness, impulsivity, and sensation seeking. However, these traits are often self-reported, which means that they are based on subjective assessments of hypothetical constructs. As a result, it is challenging to establish a direct causal relationship between personality traits and eating pathology [56].

#### 3.2.1. Perfectionism

Perfectionism is a personality trait characterized by setting unrealistic standards and striving for excellence while experiencing criticism from both internal and external sources [57]. Scientists have measured perfectionism in multiple dimensions, including self-oriented and socially prescribed perfectionism using scales such as the Frost Multidimensional Perfectionism Scale [57] and the Hewitt Multidimensional Perfectionism Scale [58]. Research has shown that AN restrictive type (ANR) [59,60] (See Table 2) and binge-eating disorder (BED) [61] are associated with perfectionism on the Hewitt Multidimensional Perfectionism Scale, while BN is associated with perfectionism on the Frost Multidimensional Perfectionism Scale [56,57].

An individual’s tend to have high scores on maladaptive and achievement-striving subscales, while BN individuals have high scores on maladaptive subscales similar to those of AN individuals. However, the findings for patients with BED have been inconsistent across studies, possibly due to different measurement methods [61,63]. Despite the consistent research on the association between eating pathology and perfectionism, doubts remain due to the relatively homogeneous sample of female subjects and the combination of different personality traits [73]. In summary, perfectionism has been found to be associated with AN, BN, and BED (See Table 3).

#### 3.2.2. Obsessive-Compulsiveness

Researchers have studied several sub-traits of obsessive-compulsiveness, such as those measured by the Maudsley Obsessional-Compulsive Inventory [83]. These include checking and washing compulsions, slowness, and doubting. One study examined female patients with obsessive-compulsive disorder (OCD), AN, or BN and found that higher scores from the Maudsley Obsessional-Compulsive Inventory were associated with eating pathology. Another study focused on a large sample of AN patients and used the Yale-Brown Obsessive-Compulsive Scale [84] to identify high percentages of obsessions and compulsions among ANR and AN binge/purge (ANBP) type patients. Obsessions included aggressiveness, contamination, somatic, and symmetry preoccupations, while compulsions included cleaning, checking, repeating, and ordering.

There has been limited research conducted to investigate the association between obsessive-compulsiveness and BN due to the different conceptualized characteristics of AN and BN (compulsive vs. impulsive) [85]. One study investigated both AN and BN subjects and found that compulsive exercise was correlated with obsessive-compulsiveness, but was only associated with ANBP, suggesting that BN was not associated with obsessive-compulsiveness [65]. However, there is still evidence of an association between BN and obsessive-compulsiveness, and the findings are mixed (Table 3) [66].

#### 3.2.3. Impulsivity

Impulsivity is a trait characterized by acting without deliberation and failing to consider risks and consequences before acting. Many psychiatric disorders, including BN, are associated with symptoms of impulse control [68]. BN patients tend to have higher levels of impulsivity than AN patients, and those defined as “multi-impulsive” show poorer response to treatment, suggesting impulsivity shapes the presentation of eating pathology [68]. Both BN patients and patients with ANBP also have higher impulsivity levels than patients with ANR and healthy controls [70], which may be due to the shared characteristic of binging, which is often induced by impulsivity. Patients with ANR also show higher impulsivity levels than healthy controls [69,70] (Table 3). Negative urgency, one of the fundamental facets of impulsivity [86,87], is the tendency to act impulsively when experiencing negative affect and is an important risk factor for binge eating [88,89,90]. Negative urgency is prospectively associated with increased binge eating, particularly among adolescent and college females [91,92]. Further, negative urgency uniquely related with binge eating when taking general negative affect into account, suggesting independent pathways for binge-eating onset [93]. Thus, it is crucial to consider impulsivity as a moderator in the association between sex hormones and eating pathology, particularly the ovarian hormone–binge eating link.

#### 3.2.4. Sensation Seeking

Sensation seekers are people who crave new and exciting experiences to maintain an optimal level of arousal. They need a higher level of stimulation than others, and repetitive experiences can make them feel bored [81]. Studies consistently show that sensation seeking is associated with binging/purging. Using the Sensation Seeking Scale invented by Zuckerman et al. [81], researchers found that BN individuals, regardless of subtype, scored higher on sensation seeking than healthy controls [71]. On the other hand, individuals with ANR had lower scores on sensation seeking compared to healthy controls [71]. Another study on females using revised Sensation Seeking Scale found that those with BN and BED had higher scores on sensation seeking than those without binging/purging [72] (Table 3).

### 3.3. Association between Sex Hormones and Personality Related to Eating Pathology

As far as we know, no study has examined the relationship between sex hormones and perfectionism. However, sex hormones may be related to other personality traits associated with eating pathologies. It is worth noting that researchers sometimes use the terms aggressiveness and impulsivity interchangeably, even though they are separate constructs that are correlated with each other [85,94,95]. In this review, we aim to demonstrate the similarity between aggressiveness and impulsivity regarding their associations with eating pathology, given their correlation.

#### 3.3.1. Androgen

Researchers have dedicated their efforts to understanding the robust personality differences between men and women. In the Eysenck’s three-dimensional model [96], testosterone has been associated with unprovoked aggressiveness in antisocial males, as found by Zuckerman [97]. Studies have also shown associations between testosterone and personalities that are prone to eating pathology among women. Some studies have implied the effect of prenatal hormone transfer in opposite-sex twins in both animal models and humans [98,99]. A later study compared the personalities of healthy opposite-sex and same-sex twins and investigated whether testosterone contributed to the personality differences. The results showed that testosterone was not systematically related to the differences in personalities [18]. However, a sex difference in proneness to aggression was observed, with women who had an opposite-sex twin exhibiting higher aggressiveness than those with a same-sex cotwin [18]. In Saudi women, higher circulating testosterone was associated with personalities such as impulsivity and disinhibition [100]. Additionally, Mathews et al. [79] (Table 3) hypothesized that congenital adrenal hyperplasia (CAH), a condition characterized by increased neurodevelopmental exposure to prenatal androgen, would masculinize female-typical personality traits. They found that females with CAH had greater physical aggression, as measured by the Reinisch Aggression Inventory, relative to female controls [75].

Research has examined the relationship between sex hormones and personality using a neuroendocrine approach. Studies have found that scores for novelty seeking on the Temperament and Character Inventory [77] and sensation seeking on the Visual Analogue Scale [78] were positively linked to norepinephrine and norepinephrine-dependent testosterone [76]. In summary, previous research has demonstrated that androgens were positively related to personality traits that reflect reward sensitivity and aggression (Table 2).

#### 3.3.2. Ovarian Hormones

While androgens have been extensively studied in relation to personality, ovarian hormones also appear to have an influence on an individual’s personality development. For example, using a facet of the Sensation Seeking Scale [101], Daitzman and Zuckerman [80] found that estradiol was positively correlated with sensation seeking and impulsivity, and loaded on the social deviancy factor. Progesterone, on the other hand, has been shown to be positively associated with binge eating as well as personalities at risk of developing other eating pathologies. Some research suggests that progesterone may be related to obsessive-compulsive behavior among females [82]. Although direct studies on the relationship between progesterone and personality are limited, they can provide insight into such associations among females (Table 2).

### 3.4. Personality as Mediator or Moderator of Sex Hormones and Eating Pathology

While previous research has explored the moderating influence of personality on sex hormones and eating pathologies, it appears that no studies have specifically examined its mediating effect. Understanding this mediating effect is crucial for establishing the causal relationship in the neurodevelopment of EDs. One study by Racine et al. hypothesized that negative urgency moderates within-person associations between ovarian hormones and emotional eating symptoms, but they found no significant effect [102]. More recently, Mikhail et al. found that trait negative affect moderated associations between ovarian hormones and disordered eating in females [23]. Specifically, they found that high trait negative affect in combination with high progesterone levels was a significant risk factor for women with a history of binge-eating episodes. However, both studies were conducted on female samples, leaving a gap in our understanding of the moderation effect of personality in the association between sex hormones and eating pathology among males.

### 3.5. Neural Representation of Personalities and Corresponding Hormone Receptor Density

As research resources and technologies have advanced, scientists have been able to investigate human characteristics and behaviors through a functional perspective by using tools such as functional magnetic resonance imaging and functional near-infrared spectroscopy to examine brain–behavior neuromechanisms. Until the 21st century, Gray’s reinforcement sensitivity theory [5] was the most widely studied and referenced neurobiological model of personality, which has since been revised by subsequent scholars [6]. This model has provided a foundation for investigating the neural correlates of personality. In this review, we aim to gather evidence on the density of steroid hormone receptors in brain structures related to personality. The hormone receptor densities in the developing brain can vary significantly in terms of their localization and pattern. Such differences could potentially alter the steroid sensitivity of the neural correlates, which might then moderate the associations between sex hormones and personality. By examining the corresponding hormone receptor density in these neural correlates, we can gain insights into the underlying neuromechanism and functional significance of the moderation effect of personality from a neuroscience perspective (See Table 4).

#### 3.5.1. Perfectionism

Neuroscientists have identified neural correlates of perfectionism, although the findings are limited. Barke et al. conducted a digit-flanker task to identify brain activity in perfectionists [103] (Table 4). Results showed activity in the medial frontal gyrus, including the anterior cingulate cortex (ACC). However, brain activity varied depending on the subtypes of perfectionism. Those with high Evaluative Concerns Perfectionism and low Personal Standards Perfectionism showed the highest activity in that area, whereas those with low Evaluative Concerns Perfectionism and high Personal Standards Perfectionism showed the lowest activity. Another study found that the components of maladaptive perfectionism (concerns over mistakes and doubts about actions) from the Chinese Frost Multidimensional Perfectionism Scale were positively correlated with gray matter volume in the ACC [114] (Table 4).

#### 3.5.2. Obsessive-Compulsiveness

Direct studies on the neural correlates of obsessive-compulsiveness are relatively limited, with most research focused on clinical samples such as patients diagnosed with OCD or obsessive-compulsive personality disorder (OCPD)—a condition characterized by inflexibility and preoccupation. Alonso et al. found a discrepancy in anterior temporal lobe volume between individuals with obsessive-compulsiveness-related dysfunctional beliefs and healthy controls, while another study comparing the amplitude of low-frequency fluctuations between patients with OCPD and healthy controls found an increased amplitude of low-frequency fluctuations in the bilateral caudate, left precuneus, left insula, and left medial superior frontal gyrus [104] (Table 4). A comprehensive review summarized the brain regions associated with OCD and found increased activities in the orbitofrontal cortex (OFC), ACC, and caudate nucleus among patients diagnosed with OCD [115]. These findings suggest an overlap between the neural correlates associated with obsessive-compulsiveness as a personality trait and OCPD and OCD as clinical disorders (Table 4).

#### 3.5.3. Impulsivity

Impulsivity is often associated with reward sensitivity, leading researchers to explore the relationship between self-control and dopaminergic (DA) brain circuits, such as the VS, VTA, and PFC. Studies have shown that the VS is correlated with impulsivity [99,100], and the nucleus accumbens (NAcc), a component of the VTA, is involved in decision-making [105]. Furthermore, other cortical areas such as the sub-thalamic nucleus (STN) have been linked to impulsivity, as lesions in the STN lead to motor impulses [108]. Interestingly, an increased amplitude of low-frequency fluctuations in the ACC and medial PFC (mPFC) has been found to be positively associated with impulsivity [110,112]. This suggests that impulsivity relates to various brain areas, including regions rich in DA neurotransmission, and highlights the relationship between impulse control, reward systems, and related brain structures (Table 4).

#### 3.5.4. Sensation Seeking

Due to its strong correlation with impulsivity, researchers have hypothesized that sensation seeking shares similar brain activity patterns with impulsivity. High sensation seekers have been shown to exhibit increased activity in the PFC and bilateral insula when facing rewards, along with insensitivity to the absence of rewards, which suggests a potential disregard for negative consequences [116]. Abler et al. confirmed this hypothesis by demonstrating a positive correlation between sensation seeking and the blood-oxygen-level-dependent (BOLD) response in the VTA and other brain areas associated with arousal and stimuli, such as the right insula and orbitofrontal cortex (OFC) [105]. However, contradictory findings have been reported. In a study that used classical appetitive conditioning, high sensation seekers showed decrease BOLD response in the NAcc, insula, and amygdala at the earlier phase, indicating blunted interoceptive processing and associative learning. In the late phase, individuals with high levels of sensation-seeking demonstrated decreased activity in the dorsal ACC, indicating blunted processing of outcome expectancy. These findings suggest an inverse relationship between sensation seeking and reward learning, which may be due to the level of risk involved in the tasks presented. These mixed findings support Zuckerman’s optimal stimulation theory, which suggests that different levels of sensation seeking require specific types of stimulation [117]. Nonetheless, it is still possible to argue that sensation seeking is positively associated with increased activity in DA-ergic areas that contribute to cognitive and emotional functions.

### 3.6. Brain Sex Hormone Receptor Density, Personality, and ED Symptomology

This section focuses on personality traits that could moderate the link between sex hormones and EDs. Specifically, we investigated how impulsivity might moderate the relationship between androgen and binge-eating-related pathology. We examined androgen receptor (AR) density in brain areas such as the VTA and NAcc. Studies in adult male rats found that testosterone was synthesized within the mesocorticolimbic system [118] (See Table 5). Additionally, a comprehensive review provided evidence that androgen and its receptors were expressed in the mesocorticolimbic DA system [119], including the VTA, NAcc, mPFC, and OFC, which were shown to program impulsivity. Although AR expression was lower than in the hypothalamus, it was present in the mesocorticolimbic DA system in male and female rodents, nonhuman primates, and humans. ARs modulate executive function in these areas (Table 5). This body of work pinpoints several potential neuromechanisms: (1). High concentration of ARs in hypothalamic nuclei innervate mesocorticolimbic nodes, influencing DA release; (2). ARs mediate the instant, nongenomic effects of androgen in the mesocorticolimbic system; and/or (3). ARs can be synthesized via neuronal bilipid membrane/G-protein-coupled intra-cellular signaling. Overall, the mesocorticolimbic system, which involves several DA-ergic pathways, was identified to associate with ARs. Thus, heightened activity in this system could stimulate androgen synthesis among humans (females in our proposed model). This may lead to cognitive expression of impulsivity, contributing to the development of binge-eating symptoms (See Figure 1).

In addition to androgen, we also investigated the neural correlates of progesterone receptor (PR) function to explore its potential moderating role in the association between progesterone and eating pathology, specifically in relation to obsessive-compulsiveness and aggressiveness. In a study using female rabbits, PR mRNA distribution in the brain was mapped by conducting ovariectomy and estradiol treatment. The results showed high expression of PRs in areas including the preoptic area, anterior hypothalamic nuclei, hippocampus, and cerebral cortex [127]. Furthermore, this study revealed that PR expression was induced by the elevation of estradiol, consistent with prior work [129]. Notably, a study on whiptail lizards identified PRs in the VTA, amygdala, NAcc, and several hypothalamic nuclei, which are responsible for processing stimuli and reward [128]. Additional studies supported the expression of PRs in these areas by demonstrating the appearance of progesterone and its metabolite 3alpha,5alpha-THP in the midbrain, specifically the VTA, regulating motivational behaviors such as mating and social interaction-related behaviors [130,131].

In summary, we propose that PR expression is associated with brain areas related to impulsivity (Table 5). Given the significant correlation between progesterone levels and impulsivity [94,95], we hypothesize that the moderation effect of aggressiveness on the association between progesterone and eating pathology could be explained by neurochemical activity in brain areas, such as the VTA and NAcc, that are related to both PR function and aggressiveness (Figure 1).

## 4. Conclusions

Based on the evidence gathered, we propose that personality traits moderate the association between sex hormones and eating pathologies. Specifically, androgen may be associated with traits such as aggressiveness or impulsivity, which are also linked to binging and purging behaviors. Our review identified neural correlates of the association between steroid hormone receptor density and eating-pathology-related personalities, providing novel neurobiological explanations. To illustrate this moderation effect, we propose new models to investigate the association between sex hormones and eating pathology. We suggest that the association between androgen and binge-eating symptoms is moderated by trait aggressiveness or impulsivity, but we hypothesize that this moderation effect only occurs in females due to the consistent protective effect of testosterone on males throughout their neurodevelopmental stages [23]. Furthermore, based on the neural correlates of AR function and impulsivity, we propose that the moderation effect of aggressiveness or impulsivity in the association between testosterone and eating pathology may be influenced by the density of ARs expressed on DA-ergic neuronal circuits in the mesocorticolimbic system (Figure 1).

We suggest that the moderation effect of personality on the association between sex hormones and eating pathology differs depending on whether the hormones are ovarian or androgens, as the interaction between estradiol and progesterone complicates describing a clear neuromechanism. While estrogen is associated with traits such as “impulsivity” and “sensation seeking,” studies consistently show a protective effect of estrogen on females’ binge eating and other eating pathologies [33,36,40]. High levels of estradiol in the preovulatory phase were associated with less binge-/emotional-eating, while low levels of estradiol and high levels of progesterone in the postovulatory phase were associated with more binge-/emotional-eating [33,132]. Therefore, personality cannot be treated as a reasonable moderator due to the different direction of the association. However, since progesterone is associated with obsessive-compulsiveness and aggressiveness, we hypothesize that these traits may moderate the association between progesterone and binge eating among females. This association may be better explained by brain activity patterns, where exaggerated neural activity in the VTA or NAcc may increase the functional expression of PR binding, leading to increased synthesis of progesterone and manifestation of impulsivity and sensation seeking, which could contribute to neurodevelopmental patterns of binge-eating behaviors (Figure 1).

Although sensation seeking has been shown to be positively associated with estrogen and eating pathology, studies have also demonstrated an inverse relationship between estrogen and eating pathology. Therefore, it may not be an appropriate moderator due to its conflicting associations with estradiol and eating pathology. However, the complex interplay between estradiol, progesterone, and other factors that affect hormonal effects on eating pathology, such as other biological and psychological changes during different stages of the menstrual cycle or general neurodevelopmental stages (e.g., prenatal, postnatal), warrants further investigation to provide evidence for the possibility of treating personality as a moderator. Studies have shown a small-to-nonsignificant relationship between estradiol and ED symptoms in females during puberty [133,134], while moderate-to-significant associations were found in normal cycling adult women [40,48]. Therefore, participants’ neurodevelopmental stage may be a crucial factor that influences the possibility of the moderation effect of personality. Additionally, as previous moderation studies have solely utilized all-female samples, future research should explore moderation analysis in males, focusing on the effects of androgen and the unique neurodevelopmental aspects of eating pathology in men.

After considering the analyzed neuroscientific evidence, this review suggests the clinical possibility of using noninvasive brain stimulation techniques, such as transcranial magnetic stimulation and transcranial electrical stimulation, to modulate brain areas (e.g., mesocorticolimbic DA system) that appear to be sensitive to developmental exposure to sex hormones that program personality traits underlying eating pathology. To avoid inconsistent and scattered findings, future studies should also use more validated personality measurements to assess hormone:personality and personality:eating pathology relationships. Methods of assessing neural activity, such as electroencephalograms, should be implemented to confirm the proposed neurodevelopmental model underlying eating pathology. Although the review describes a new brain-based approach to understanding the etiology of EDs, most of the evidence was from association studies. Therefore, the potential underlying neurochemical mechanisms within identified brain circuits were not explored. Future research in this field would benefit from identifying receptor-specific intracellular mechanisms corresponding to the development of DA-ergic neurocircuits to corroborate the proposed model. Indeed, emerging molecular genetic data describing metabolic pathways are providing a mechanistic insight into AN as a metabotropic neuropsychiatric disease [135]. An important paradigm shift in the field of ED research.

## Figures and Tables

**Figure 1 brainsci-13-00859-f001:**
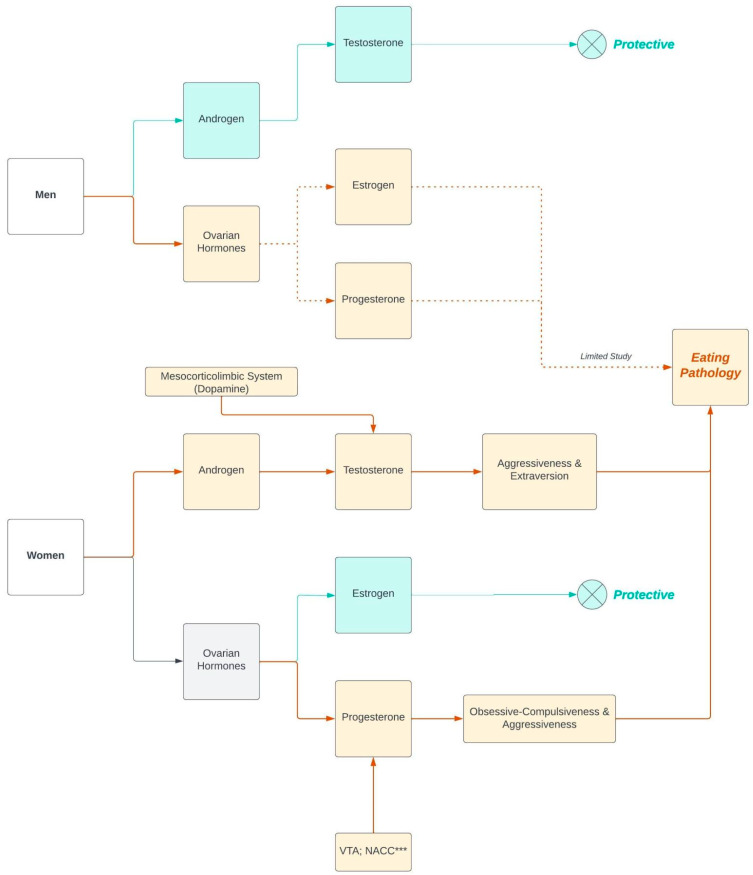
Notes: *** VTA: ventral tegmental area; NAcc: nucleus accumbens. Designed using Lucidchart software (Lucid Software Inc., South Jordan, UT, USA).

**Table 1 brainsci-13-00859-t001:** Association Between Sex Hormones and Eating Pathology.

	Anorexic Behaviors (Restrictive)	Association Indicator	Anorexic Behaviors (Binge/Purge)	Association Indicator	Bulimic Behaviors	Association Indicator	Binge Eating	Association Indicator
Androgen	[24]	male ↓	[24]	male ↓	[25]	female ↑	[17]	female ↓
	[20]	female ↑	[20]	female ↑	[22]	male and female; nonsignificant	[22]	male and female; nonsignificant
	[22]	male and female; nonsignificant	[22]	male and female; nonsignificant	[20]	female ↓	[21]	male and female; nonsignificant
	[21]	male and female; nonsignificant	[21]	male and female; nonsignificant	[19]	female ↓	[26]	male ↓
					[21]	male and female; nonsignificant	[27]	male ↓
					[28]	female ↑	[13]	male ↓
							[29]	male ↓
Ovarian Hormones	[30]	female ↑	[30]	female ↑	[30]	female ↑	[30]	female ↑
	[31]	male and female rodents; ↑	[32]	female rodent ↓	[33]	female; E (↓); PR (↑)	[34]	female rodent ↓
	[35]	male and female rodents; ↑	[36]	female; E (↓); PR (↑)	[32]	female ↓	[37]	female ↓
	[38]	↑	[39]	female; PR (↑)	[36]	female; E (↓); PR (↑)	[40]	female; E (↓); PR (↑)
					[41]	female ↓	[36]	female; E (↓); PR (↑)
	[42]	male and female rodents; ↑	[41]	female ↓	[39]	female; PR (↑)	[32]	female rodent ↓
	[43]	female rodent; ↑	[44]	female rodent ↓	[44]	female rodent ↓	[41]	female ↓
	[41]	female ↓					[39]	female; PR (↑)
	[39]	female; PR (↑)					[44]	female rodent ↓
	[44]	female rodent ↓					[45]	female; E (↑); PR (↑)

Notes: Studies that did not investigate a specific type of ED (e.g., a global eating disorder syndrome) are put into each column. Studies that investigated the symptoms are put under the categories that include the symptoms. E = estrogen; P = Progesterone; ↑ = positive association; ↓ = negative association.

**Table 2 brainsci-13-00859-t002:** Association between Personality and Eating Pathology.

	Restrictive Eating	Association Indicator	Binging/Purging	Bulimic Symptoms	Association Indicator	Binge Eating	Association Indicator
Perfectionism	[59]	male and female; ↑		[62]	female; ↑	[63]	female; ↑
	[60]	female; ↑		[64]	female; ↑	[61]	female; ↑
Obsessive-Compulsiveness	[65]	female; ↑		[66]	female; ↑		
	[67]	male and female; ↑		[67]	male and female; ↑		
Impulsivity	[68]	male and female; ↑		[68]	male and female; ↑		
	[69]	female; ↑					
	[70]	male and female; ↑					
Sensation Seeking	[71]	male and female; ↓		[71]	male and female; ↑	[72]	male and female; ↑
				[72]	male and female; ↑		

Notes: ↑ = positive association; ↓ = negative association.

**Table 3 brainsci-13-00859-t003:** Association between Sex Hormones and Personality.

	Hormonal	Scale/Inventory	Association Indicator	Physiological	Scale/Inventory	Association Indicator
Androgen	[74]	Aggression proneness, Reinisch Aggression Inventory [75]	Females; ↑	[76]	Novelty Seeking; Temperament and Character Inventory [77]; Visual Analogue Scale [78]	Males; ↑
				[79]	Physical aggression; Reinisch Aggression Inventory [75]	Females; ↑
Ovarian Hormones	[80]	1. Sensation Seeking; 2. Impulsivity; Sensation Seeking Scale [81]	Males; (1–2) ↑			
	[82]	Obsessive-Compulsive behavior; The Minnesota Multiphasic Personality Inventory [82]	Females; P (↑)			

Notes: E = estrogen; P= progesterone; ↑ = positive association; ↓ = negative association.

**Table 4 brainsci-13-00859-t004:** Neural Correlates of ED-related Personalities.

ED-Related Personalities	Neural Correlates	Association Indicator
Perfectionism	Anterior Cingulate Cortex [103]	↑
Obsessive-compulsiveness	Bilateral Caudate; left Insula; left Medial Superior Frontal Gyrus [104]	↑
Impulsivity	Nucleus Accumbens [105]	↑
	Ventral Striatum [106]	↑
	left Insula/Inferior Frontal Gyrus [107]	↑
	Sub-Thalamic Nucleus [108]	↑
	Prefrontal Cortex [109]	↑
	Anterior Cingulate Cortex; Medial Prefrontal Cortex [110]	↑
	Ventral Striatum [111]	↑
	Anterior Cingulate Cortex; Medial Prefrontal Cortex [112]	↑
Sensation seeking	Ventral Tegmental Area [105]	↑
	right Insula; Orbital frontal Cortex [113]	↑
	NAcc; Insula; Amygdala [113]	↓
	dorsal Anterior Cingulate Cortex [113]	↓

Notes: ↑ = positive association; ↓ = negative association.

**Table 5 brainsci-13-00859-t005:** Neural Correlates of Sex Hormone Receptor Density.

Sex Hormone Receptor (Density)	Neural Circuit	Association Indicator
Androgen	Mesocorticolimbic System (VTA, NAcc, mPFC, OFC) [120]	↑
	Mesocorticolimbic System (VTA, NAcc, mPFC, OFC) [121]	↑
	Mesocorticolimbic System (VTA, NAcc, mPFC, OFC) [122]	↑
	Mesocorticolimbic System (VTA, NAcc, mPFC, OFC) [123]	↑
	Mesocorticolimbic System (VTA, NAcc, mPFC, OFC) [124]	↑
	Mesocorticolimbic System (VTA, NAcc, mPFC, OFC) [125]	↑
	Mesocorticolimbic System (VTA, NAcc, mPFC, OFC) [126]	↑
	Mesocorticolimbic System (VTA, NAcc, mPFC, OFC) [118]	↑
Progesterone	Supraoptic area; Anterior Hypothalamic Nuclei; Hippocampus; Cerebral Cortex [127]	↑
	VTA, NAcc [128]	↑

Notes: ↑ = positive association; ↓ = negative association; VTA = ventral tegmental area; NAcc = nucleus accumbens; mPFC = medial prefrontal cortex; OFC = orbital frontal cortex.

## Data Availability

Not applicable.

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
