# Peer review of "Neurodevelopmental Model Explaining Associations between Sex Hormones, Personality, and Eating Pathology"

_brainsci, 2023, doi:10.3390/brainsci13060859_

Round 1
Reviewer 1 Report
This manuscript aims to summarize and synthesize the literature on sex steroid hormones, personality, and eating pathology. The authors conclude that aggressiveness, impulsivity, and obsessive-compulsiveness may impact (via mediation or moderation) relationships between sex hormones and eating pathology, but only among females, and suggest that sex hormone receptor density in the mesocorticolimbic pathways may underlie (at the neural level) such associations. While this is an interesting and important line of work, the manuscript could be condensed and there are a number of issues that need to be addressed for accuracy and clarity:
Broad Issues:
1) Terminology: The manuscript reads as though much of the work on hormones and eating pathology has been done in eating disorder samples and/or has been shown to predict the development of a clinical eating disorder. However, given that hormones are altered secondary to eating disorders (during the ill state), much of the etiologic-based research has used non-clinical/population-based samples and have examined disordered eating symptoms (dimensionally, across spectrum of severity). It will be important to update the manuscript for accuracy, and it would likely be beneficial to explicitly acknowledge the fact that studies have tended to explore symptoms rather than clinical eating disorders.
2) Organization of content: I would encourage the authors to revise the manuscript to improve the organization of content. There are several places where related findings are not clustered together, which hinders flow and interpretation. For example, a finding is mentioned, and then additional sentences switch to reviewing other findings and then switch back to similar content/findings. One example of this is within the first paragraph where it is noted that prenatal exposure to testosterone contributes to disordered eating among different-sex twins, then content shifts to connections between sex hormones and genetics and genetic risk for eating pathology is mentioned, and then the paragraph switches back to discussing early testosterone exposure. Additionally, the introduction could likely be condensed and just focused on outlining the purpose/aims of the review. Much of the content discussed within the introduction section is then discussed again under the literature review.
3) Avoid overstating findings: Relatively few studies on this topic area have been conducted and much of the human work has been correlational and/or has relied on indirect methods. I would encourage the authors to do a careful read of the manuscript and to be more tentative in their language and conclusions; along these lines, avoid using words like “proven.” It may also be beneficial to more explicitly acknowledge findings that do have stronger evidence (e.g., replication of effects has been obtained; corroborating data from animal and human work) versus initial/preliminary evidence.
4) Clarity: Several concepts, ideas, methodological approaches, etc. are raised throughout the manuscript but tend to be mentioned in a very brief manner without providing the reader with sufficient information for interpretation. For example, organizational hormone effects, non-shared environment, various personality constructs (like negative urgency), mediation vs moderation, androgen receptor CAG repeats are mentioned but never defined or explained. Similarly, an emphasis is placed on density of steroid hormone receptors, but it is unclear what this would represent and/or exactly why density of steroid hormone receptors might be key (related to hormone levels? function? Etc). Pubertal timing is also mentioned but there is no information as to how this might connect to hormone levels, etc.
5) Narrowing the focus and reducing manuscript length: The manuscript would benefit from a clearer and narrower focus. For example, several personality constructs are discussed but many of these are not considered “key” to eating pathology and/or the personality constructs discussed in relation to eating pathology are not the same ones discussed in relation to hormones or neural regions, etc. I would encourage the authors to pull from more recent reviews/meta-analyses and focus only the main personality characteristics relevant to eating pathology. This will result in a substantial amount of cutting but will also improve readability and overall interpretation of the work.
Other Comments:
1. There are additional search terms that I was expecting given the topic and types of research conducted (often non-clinical samples or animal studies that focus on behavioral outcomes not disorders), e.g., gonadal hormones, estrogen/estradiol, testosterone, androgen, disordered eating, eating pathology, or even symptom level key words like binge eating, emotional eating, etc.
2. When discussing hormone effects, be clear about developmental life stage (prenatal/perinatal, puberty, post-puberty or young adulthood, mid-to-late life) since timing of hormone exposure matters for many outcomes, including eating pathology.
3. In regard to the inverse association between testosterone and AN in males, it would be important to consider whether the effect is merely secondary to illness. If levels are low even after recovery, which I believe there is some evidence for (e.g., remaining at the low end of normal), then that could be noted and explained in terms of potentially having more etiologic relevance and not just being a physiological response to low weight and nutritional status. Similarly, the authors comment that there is a gap in knowledge on ovarian hormones and AN, but within that discussion, it seems important to speak to the fact that the gap is largely in terms of determining whether there are etiologic links (not just disruptions/low ovarian hormones secondary to illness). It may also be worth providing some speculative hypothesis – since low estradiol seems risky for binge eating, would you anticipate that higher estradiol might be related to more restrictive/undereating behavior (typical of AN)?
4. It may be best to discuss hormone effects on feeding behaviors prior to dividing data into animal versus human models OR be careful in the placement of that text, which is provided under human studies but is largely derived from experimental animal work.
5. The text reads as though the authors suspect that personality may have mediation effects in the relationship between hormones and eating pathology, more so than moderation effects. This is in contrast to the prior studies (the few studies that have considered personality effects in relation to hormones and eating), in which moderation was tested. It would be beneficial to include some explanation as to why investigators may have focused on moderation rather than mediation as well as why you think mediation could be at play. Conceptually, I understand the moderation effects to likely be derived from the fact that sex hormones influence eating behavior (and these prior studies focused on dysregulated eating outcomes) and that personality may serve as between-person risk factors that modifies vulnerability to hormonal effects on dysregulated eating outcomes. However, maybe mediation effects would be more relevant for eating disorder symptoms that aren’t eating/food based, and presumably, outcomes where sex hormones may have less of a direct effect? This point also relates to a broader theme that isn’t completely clear in the manuscript – hormone and personality effects likely vary based on symptom/outcome so it is unlikely that one theoretical model/approach would apply to all forms of eating pathology outcomes/symptoms.
6. I would encourage the authors to reduce the number of acronyms used in the manuscript.
7. In the discussion of personality as a mediator or moderator of sex hormones and eating disorders, the authors conclude that the studies were conducted on female samples “indicating a decreased representation of the general population.” It is a bit unclear what the authors are suggesting could have improved this prior work. I suspect the studies focused on females because they studied ovarian hormone effects across the menstrual cycle. The authors of this review might be suggesting that it would be important to explore whether personality may moderate ovarian hormones effects (or other sex hormones?) in other genders but that isn't clear.
1. The following content needs to be revised/reworded: "Mikhail et al. showed that personality moderates hormone-ED association. They found that a negative effect interacts with low estradiol or high progesterone, contributing to the risk of females developing eating disorders." Rather than "that a negative effect interacts," I believe it should read "that negative affect interacts..."
Author Response
Please see the attachment, thank you!

Reviewer 2 Report
The authors have reviewed a Novel Neurodevelopmental Model Explaining Associations Between Eating Pathology, Personality, and Steroid Hormones,
The authors have to make at least 4 figures (see an example https://pubmed.ncbi.nlm.nih.gov/22567385/) to show how neurodevelopment model explains these phenomenon. Without that it looks irritating to read so much. Figures should explain the details of the review. The authors can add the post translational modifications like described in below review (https://link.springer.com/article/10.1007/s00726-021-03023-6
) and role of some kinases (Check figure 1 for list of some important kinases https://www.nature.com/articles/s41589-018-0194-1) in related to the above phenomenons described.
The authors can see an example how figures explains the review.
https://link.springer.com/article/10.1007/s00726-021-03023-6
https://pubmed.ncbi.nlm.nih.gov/22567385/
looks ok
Author Response
Please see the attachment, thank you!

Reviewer 3 Report
Zhao and Gobrogge present a novel model that characterizes the associations between eating disorders, personality, and steroid (sex) hormones. The authors conducted a comprehensive review of literature across several databases to integrate findings from discrete studies that exemplify the current understanding of the above associations. The authors highlight the mediation and moderation of personality traits on sex hormone (androgen, E2, and progesterone) induction of eating disorders in males and females. Due to limited preclinical and clinical research at the intersection of personality, SH, and ED, it is complicated to clearly unravel the above associations. Yet, the authors seamlessly discussed mixed findings and propose sex-specific personality-trait/behavior models that facilitate SH moderation of ED as substantiated by neural correlative investigations. The proposed association suggests the possibility of the use of non-invasive neural stimulation interventions in the treatment of personality trait-ED. While requiring concrete studies to support the author’s hypothesis, this review article in the present form is of appropriate scope, rigor, and comprehension. Below are minor comments.
Minor comments:
1. Can you be consistent in the use of terms “sex hormones” vs “steroid hormones”. Consider using sex hormones rather than steroid hormones (cortisol, sex hormones) as appropriate. Particularly because you discuss the neurobiological networks that are associated with personality including the FFFS and BIS (high activity) involvement, which likely also involves upregulated expression/secretion of glucocorticoids.
General comment:
2. Ensure consistency in text font throughout the document.
3. For tables 4 and 5, are the E and P notations missing? The figure legends

Author Response
Please see the attachment, thank you!

Round 2
Reviewer 1 Report
The authors have carefully considered prior comments and thoughtfully addressed the concerns. The manuscript is improved, and I have only a few remaining comments:
1 1. Given that Negative Urgency is the leading facet of impulsivity that has been linked to eating pathology, particularly binge eating symptomatology, I would recommend addressing/integrating this in the impulsivity section. This will also help provide context as to why negative urgency was a key personality component that has been investigated as a moderator of ovarian hormone-binge eating relationships.
2. The direction of estrogen (E) effects on eating pathology outcomes is certainly complicated given that the interaction between E and progesterone seems to be particularly important for phenotypic expression of binge eating symptoms. One major part of this, which would be useful to clarify in the manuscript, is that progesterone antagonizes E effects (likely is resulting in low E action). Low estradiol has also been shown to moderate genetic effects on binge eating in girls during puberty (notably, these genetic moderation effects are present during puberty/adolescence but phenotypic ovarian hormone relationships with binge eating do not seem to appear until later in development). I point this out because I did get stuck on the suggested link between lower E and higher levels of restrictive eating. Moreover, in female rodents, removal of E and P leads to increase food intake and binge eating behavior, and administration of E reverses the effect (reduces intake). It certainly remains an empirical question as to what the direction of effect might be in AN, which is also complicated by the fact that in the ill-state women with AN would have low ovarian hormones. I encourage the authors to consider this a bit more.
3. The figure shows the outcome as “eating disorder.” Based on the other changes to the manuscript, this should be updated to “eating pathology.”
Author Response
Please see the attached reply. Thank you!

Reviewer 2 Report
I think graphic representation must be there for review. Otherwise its not interesting to read by readers. The authors have not addressed much as suggested by reviewers.
its ok
Author Response

(The authors gave the same response as above.)
